# Network-Based Assessment of Minimal Change Disease Identifies Glomerular Response to IL-7 and IL-12 Pathways Activation as Innovative Treatment Target

**DOI:** 10.3390/biomedicines11010226

**Published:** 2023-01-16

**Authors:** Øystein Eikrem, Bjørnar Lillefosse, Nicolas Delaleu, Philipp Strauss, Tarig Osman, Bjørn Egil Vikse, Hanna Debiec, Pierre Ronco, Miroslav Sekulic, Even Koch, Jessica Furriol, Sabine Maria Leh, Hans-Peter Marti

**Affiliations:** 1Department of Clinical Medicine, University of Bergen, 5020 Bergen, Norway; 2Department of Medicine, Haukeland University Hospital, 5021 Bergen, Norway; 32cSysBioMed, 6646 Contra, Switzerland; 4INSERM UMRS 1155, Hôpital Tenon, 75020 Paris, France; 5Department of Pathology and Cell Biology, Columbia University, New York, NY 10032, USA; 6Department of Pathology, Haukeland University Hospital, 5021 Bergen, Norway

**Keywords:** minimal change disease, membranous nephropathy, transcriptome, IL-7, IL-12

## Abstract

**Background:** Minimal change disease (MCD), a major cause of nephrotic syndrome, is usually treated by corticosteroid administration. MCD unresponsiveness to therapy and recurrences are nonetheless frequently observed, particularly in adults. To explore MCD-related pathogenetic mechanisms and to identify novel drug targets ultimately contributing to novel therapeutic avenues with a certain specificity for MCD, we compared glomerular transcriptomes from MCD with membranous nephropathy (MN) patients and healthy controls. **Methods:** Renal biopsies from adult patients with MCD (n = 14) or MN (n = 12), and non-diseased controls (n = 8) were selected from the Norwegian Kidney Biopsy Registry. RNA for 75 base-pair paired-end RNASeq were obtained from laser capture micro-dissected (LCM) glomeruli from FFPE sections. Transcriptional landscapes were computed by combining pathway-centered analyses and network science methodologies that integrate multiple bioinformatics resources. **Results:** Compared to normal glomeruli, cells from MCD displayed an inflammatory signature apparently governed by the IL1 and IL7 systems. While enrichment of IL1 production and secretion was a shared feature of MCD and MN compared to normal tissue, responses involving IL7 pathway activation were unique to MCD. Indeed, IL7R expressed by glomeruli was the most upregulated gene of the interleukin family in MCD versus normal controls. IL7 pathway activation was paralleled by significant enrichment in adaptive immune system processes and transcriptional regulation and depletion in pathways related to energy metabolism and transcription. Downregulation of these organ function-related themes again occurred predominately in MCD and was significantly less pronounced in MN. Immunofluorescence and immunohistochemistry, respectively, confirmed the expression of phosphorylated IL-7 receptor alpha (IL7RA, CD127) and IL12 receptor beta 1 (IL12RB1) proteins. **Conclusions:** Gene expression profiling of archival FFPE-biopsies identifies MCD-specific signatures with IL7RA and IL12RB1 as novel targets for MCD treatment.

## 1. Introduction

Minimal change disease (MCD) reflects a very common cause of idiopathic nephrotic syndrome (INS). It usually presents with proteinuria, hyperlipidemia, edema and hypoalbuminemia as the most common symptoms, and it accounts for 70–90% of all cases of INS in children >1 year old and for 10–15% in adults [1]. Frequent relapses will eventually lead to decreasing kidney function.

Response to steroid treatment represents a central prognostic factor, as unresponsive patients will eventually evolve towards chronic kidney disease (CKD). Unlike other glomerular diseases, MCD is usually characterized by a responsiveness with complete remission of proteinuria following the initiation of corticosteroid treatment [2,3]. However, previous studies [2,3] have underlined that response in children is faster than in adults and that adults have a higher probability of relapse than children.

Although pathogenetic mechanisms of MCD are still, by and large, unclear, a number of studies consistently indicate that immune system dysregulation(s) do play a major role [1]. This may be particularly relevant regarding the induction of podocyte alterations affecting glomerular filtration. Early studies suggested a T-cell involvement based on clinical observations [4], such as measles-induced remission of MCD, and the association of relapse with decreased levels of regulatory T-cells [5]. Moreover, a role of B-cells has emerged, especially related to the reported effectiveness of rituximab in MCD treatment [6,7].

On the other hand, intriguingly, a higher frequency of MCD has been observed in patients with hematologic malignancies and, in particular, Hodgkin’s lymphoma [8].

In the past, it has been proposed that ill-defined glomerular permeability factor(s) produced by B- or T-cells might be involved in MCD pathogenesis [9] and different cytokines have been suggested to play this role. Interleukin 13 (IL-13) was shown to stimulate podocyte protein trafficking and proteolysis in vitro [10], and overexpression of IL-13 in rats was found to induce an MCD-like nephropathy [11]. However, other cytokines, such as IL-2, IL-4, IL-8, IL-18, and TNF-α, have also been suggested as possible driving forces in MCD pathogenesis [12,13]. In particular, expression of IL-7 receptor (IL-7R) was reported to be induced in murine podocytes in adriamycin (ADR)-induced mouse nephropathy [14]. Notably, however, data regarding human disease is scarce.

Omics-related technologies are providing novel insights into the pathogenetic background of human disease. Most importantly, their high potential for the analysis of kidney diseases is emerging [15]. However, there is a paucity of clinical studies [16,17,18], and in particular, MCD has not been investigated to a great extent. To fill this knowledge gap, here we investigated transcriptional profiles of microdissected glomeruli from kidney biopsies from patients with MCD by RNA sequencing (RNASeq) and compared them with those obtained from non-diseased controls and primary membraneous nephropathy (MN) patients. For our studies, we have used archival, formalin-fixed and paraffin-embedded (FFPE) kidney biopsies. Although FFPE material is not ideal compared to fresh tissues, we have previously demonstrated a close concordance of RNAseq data between FFPE and fresh kidney tissues [19]. Establishing and refining such methodology is indeed key to enabling FFPE material, often representing invaluable disease phenotypes and decades of documented treatment responses observed in clinics, to be used in studies in the field of enabling precision medicine based on omics technologies.

## 2. Patients, Materials and Methods

### 2.1. Patients

Renal biopsies from largely untreated adult patients with MCD (n = 14, females n = 6, males n = 8; age: 50 ± 16 years; eGFR 90 ± 29 mL/min/1.73 m^2^; proteinuria 8.7 ± 3.7 g/d), with primary membranous nephropathy (MN; PLA2R positive: n = 6, females n = 2, males n = 4, age 58 ± 16, eGFR 90 ± 14 mL/min/1.73 m^2^, proteinuria 3.4 ± 4.4 g/d; and PLA2R negative: n = 6, females n = 4, males n = 2, age 53 ± 18, eGFR 85 ± 18 mL/min/1.73 m^2^, proteinuria 2.4 ± 2.6 g/d), as well as with normal kidney histology (N_CTRL; n = 8, females n = 4, males n = 4, age 27 ± 11 years, eGFR 113 ± 25 mL/min/1.73 m^2^, proteinuria 0.2 ± 0.2 g/d), were selected from the Norwegian Kidney Biopsy Registry. Other primary or systemic kidney diseases, such as diabetic and/or hypertensive nephropathy or vasculitis were exclusion criteria. Among the MN patient group, 1/12 had concurrent diabetes type 2, 10/12 subjects displayed coexisting, largely ACEI or ARB (angiotensin receptor blocker) treated, hypertension, and 3/12 had already received corticosteroid-based immunosuppression. Among the MCD subjects, 3/14 had untreated hypertension, and none were on documented immunosuppressive treatment at the time of biopsy, respectively; 2/14 had a history of smoking, and none had diabetes. All controls were untreated.

Due to the paucity of non-diseased kidney biopsies, control samples from patients younger on average than those with MCD and MN had to be used. In all cases, informed consent was provided.

### 2.2. RNA Purification and Sequencing

Formalin-fixed and paraffin-embedded (FFPE) sections of stored kidney biopsy tissue were microdissected, as previously reported [16], to obtain glomerular cross-sections. RNA was extracted and purified from enzymatically digested specimens using the High Pure FFPE RNA Isolation Kit^®^ (Roche Molecular Systems, Inc., Pleasanton, CA, USA) according to previously described techniques [20]. RNA libraries were generated using the TruSeq RNA Access kit (Illumina Inc.,San Diego, CA, USA) and sequenced (75 base-pair paired-end RNAseq) according to protocols previously utilized by our group [21,22]. Gene expression data are available in the GEO repository through accession number GSE216841.

### 2.3. Processing Gene Expression Data Prior to Mapping Transcriptional Landscapes

Based on unsupervised clustering and PCA correlation analysis, potential batch effects within the RNASeq data were mitigated using Surrogate Variable Analysis (SVA) correction in combination with counts per million (CPM) normalization. Subsequently, using a standard DESeq2 workflow, including shrinkage of log2 fold changes (slfc) via the adaptive shrinkage estimator from the ashr package, differential gene expression was computed to compare all groups [23]. The resulting lists comprising 17,561 genes passing all criteria set by the analyses described below were subsequently ranked based on slfc.

### 2.4. Mapping Transcriptional Landscapes

To avoid introducing arbitrary significance thresholds prior to the contextualization of changes in gene expression as a whole, a gene set (GS)-based approach to analyzing these data was chosen [24]. In this context, a GS is an a priori-defined group of genes annotated to reflect one specific trait that its members share [24].

A total of 26,251 GSs compiled, curated and updated by research consortia, such as the Gene Ontology project and Reactome were interrogated. The method for assembling this GS collection is described in [25]. For this study, the resources were compiled on 1 March 2019 and downloaded from http://download.baderlab.org/EM_Genesets/ (accessed on 15 March 2019) were they are archived as well.

Of the 26,251 GSs, 10,325 passed the default size thresholds of >10 and <500 member genes, therewith mitigating the limitations of gene set enrichment analyses (gsea) with respect to very small or very large GSs. Subsequently, the ranked lists obtained as described above were interrogated for significant enrichment or depletion patterns formed by the 10,325 GSs’ member genes. Specifically, fast gene set enrichment analyses (fgsea) [26], a speed-optimized variant of the original gene set enrichment analysis algorithm [24], were used for that purpose.

To comprehensively interpret the results generated via fgsea, they were mapped using concepts and algorithms, as described in detail by us previously [27,28,29,30]. Briefly, any GS yielding a significant enrichment or depletion signal (adjusted *p*-value < 0.001), with at least 33.34% of its member genes comprised in its leading edge (LE) when comparing MCD vs. N_CTRL was mapped as a node within the basic network. The latter is organized by forces of attraction (edge-weights) proportional to pair-wise overlap in gene set member genes in case their LEs’ shared at least 5% of their member genes when comparing MCD vs. N_CTRL.

Markov cluster algorithm (MCL), considering the edge-weights and all other settings set on default, was applied to identify clusters within the network. In technical terms, the clusters identified by MCL are identified via simulating the stochastic flow within the network [31]. The resulting clusters were then interpreted in detail, based on the gene sets’ names, fgsea-related statistics, as well as distribution and function of their LE-genes, to infer a biological theme from each cluster. The final visualization was achieved using the yFiles organic layout algorithm in Cytoscape 3.7.1 [32]. Subsequently, results from the other comparisons relevant here (MCD vs. MN and MN vs. N_CTRL) were also mapped on this initial network structure.

For individual genes, when members of GSs pass all significance thresholds listed above, an adjusted *p*-value of <0.05 and a shrunk fold change criterion of >2 for upregulation and <−2 for downregulation were required to be considered significant.

### 2.5. Interleukin Interactome

A comprehensive map of the interleukin ligand/receptor interactome was derived based on Hugo gene nomenclature committee (HGNC) families 42, 43, 33, 5, on published literature reviews [33,34,35,36,37,38], and on String DB interactions for cytokines with no established binding partners. This interactome was imported into Cytoscape 3.7.1 [32], and a yFiles orthogonal layout was applied. Subsequently, all cytokine ligand/receptor data were mapped onto this network for comparative evaluation.

### 2.6. Immunohistochemistry

To optimize immunohistochemical (IHC) staining protocols, several different antibodies were used on test tissues, and Proteinatlas (Proteinatlas.com) recommendations were carefully followed. Finally, IHC was used for IL-12 receptor beta 2 (IL12RB2) and human AXL detection, and immunofluorescence (IF) for IL-7 receptor alpha (CD127).

For IL12RB2, we used an antibody from Sigma Aldrich (Burlington, MA, USA) cat.no HPA024168 with a dilution of 1:1000 for primary antibody, and an incubation time of 1 h in a humidity chamber.

For AXL detection, we used an Invitrogen 7E10 monoclonal antibody from Thermofisher (Waltham, MA, USA) cat. no MA5-15504, with a dilution of 1:10,000 and 1 h incubation time in a humidity chamber.

For the detection of IL-7 receptor alpha (CD127), we used immunofluorescence due to a lack of signaling by using standard IHC on tonsil test tissues. After additional test-rounds, we used an antibody from Abcam (Cambridge, UK) cat. no. ab118527 recognizing phosphorylated CD127 with a dilution of 1:100 and incubation time of 1 h in a humidity chamber.

## 3. Results

### 3.1. Network-Based Delineation of MCD’s Glomerular Transcriptional Landscape

As shown in Figure 1, the top section shows clusters 1–5; an analysis of the network delineating MCD’s glomerular transcriptional landscape led to the identification of 51 GSs, e.g., pathways, significantly enriched in MCD compared to normal controls (N_CTRL). Concurrently, 95 GSs pathways appeared to be significantly depleted (Figure 1; bottom section, clusters 6–10).

By applying previously described methodologies [27,28,29,30], we could resolve redundancy via MCL-clustering and subsequent detailed interpretation of the clustered network areas considering GSs’ names, fgsea-parameters and the specific genes underlying their significant enrichment or depletion signal. On this basis, each cluster could be summarized and annotated with a term denoting the biological theme that the changes as a whole were associated with (Figure 1, Figure 2 and Figure 3).

In particular, themes upregulated in MCD vs. N_CTRL (Figure 1) did delineate a significant inflammatory component comprising activation of the pathways associated with response to IL-7 (Figure 1; Cluster 1), IL-12 and IFN-gamma (Figure 1; Cluster 2), connected to pathways pertaining to the modulation of adaptive immune responses.

In addition, genes governing IL-1 beta production and secretion formed an additional theme (Figure 1; Cluster 3), in turn, connected to significantly enriched GS, suggesting the presence of gene expression patterns specific for granulocyte chemotaxis (Figure 1; Cluster 4). Despite such signal, however, we could not detect significant amounts of granulocytes in our kidney biopsies, and in general, there was an expected downstream effect related to this specific type of immune activation. Lastly, the upregulation of genes related to vasculature development underlie the only enriched cluster theme apparently not directly associated with immune system activation (Figure 1; Cluster 5).

Possibly as a consequence of the inflammatory milieu described above, the five biological themes identified by clustering GSs depleted in MCD vs. N_CTRL (Figure 1; clusters 6–10) exclusively pertained to GSs associated with glomerular homeostasis and function. They include themes such as ribosome biogenesis, transcription and translation, symporter activity, cellular respiration and metabolism.

### 3.2. Specificity of Alterations Mapped within the MCD vs. N_CTRL-Focused Network

To assess the specificity of the alterations detected in MCD vs. N_CTRL, we explored the glomerular transcriptome in MN as a secondary reference group. Mapping fgsea adjusted *p*-values for MCD vs. MN (Figure 2) within the same network structure derived from MCD vs. N_CTRL (Figure 1) allowed for a direct comparison.

For the five previously observed inflammation-related GS-clusters (Figure 1 and Figure 2; clusters 1–5), a dichotomy was apparent. While the expression of the IL-1, chemotaxis and vasculature-related themes was comparable in MCD and MN (adjusted *p*-value > 0.05), clusters associated with response to IL-7 and IL-12, including adaptive immune response and regulation of effector functions showed a significant enrichment in MCD as compared to MN (Figure 2; clusters 1 and 2). GS involved at different steps of gene expression regulation and transcription were also found enriched in MCD when compared to MN glomeruli.

Importantly, a comparison of MN vs. N_CTRL transcriptomic profiles further underlined this apparent MCD-specificity, since of the 51 GSs enriched in MCD vs. N_CTRL, only two, namely *Regulation of IL-1 Beta Production* and *Regulation of IL-1 Beta Secretion*, yielded an adjusted *p*-value < 0.05 when assessed in MN vs. N_CTRL glomeruli. (Figure 3; clusters 1–5).

Regarding clusters delineating downregulated themes (Figure 1 clusters 6–10), metabolic reprogramming appeared to represent a particularly prominent feature of MCD as well. Indeed, the vast majority of these GSs, with special reference to cellular respiration, were also significantly depleted when comparing MCD with MN while being unchanged in MN vs. N_CTRL specimens (Figure 2 and Figure 3; Cluster 7).

A more detailed assessment of alterations in biological themes for all comparisons within the three study groups was obtained by plotting adjusted *p*-values per theme (Figure 4), indeed confirming that all GSs comprised in the IL-7 and IL-12-governed clusters are exclusively enriched in an MCD-context (Figure 4, *Response to IL7 and transcriptional regulation* and *IL12, IFN and regulation of lymphocyte-mediated immunity*). This is in contrast with other upregulated themes (Figure 4, biological themes 3–5). For GSs depleted in MCD vs. N_CTRL (Figure 4, biological theme 6–10), significant differential expression with MCD-specificity was apparent only for cellular respiration (Figure 4, biological theme 7).

Briefly, computation and interpretation of transcriptional landscapes provided a systematic data-driven framework suggesting that, in contrast to MN, glomerular tissue and potentially organ function of patients with MCD may be governed to an extent by IL-7 and IL-12.

### 3.3. IL-7 and IL-12-Mediated Signaling Member Genes

A more detailed assessment of all member genes belonging to the two key pathways identified via network analyses above is presented in Figure 5. Indeed, coordinated upregulation of IL-7-mediated signaling distinguished MCD compared to MN (Figure 5A,B) that, in turn, remained remarkably similar to N_CTRL (Figure 5C). Prominently involved genes encode the IL-7 receptor alpha (CD127) and Jak/Stat signal transduction pathway, in addition to several histones regulating gene expression (Figure 5A).

Regarding IL-12-signaling, genes significantly upregulated in MCD vs. the reference groups (Figure 5D–F) included, as expected, those encoding IL-12 receptor alpha and beta chains but also tyrosine kinase 2, IL-1 beta, and CCL3, a chemokine attracting myeloid cells. This pattern was absent when comparing MN vs. N_CTRL samples (Figure 5F).

### 3.4. Mapping Alterations within the Interleukin Interactome

To gain additional insights into the interleukin system as a whole and potentially activated in MCD glomeruli, we mapped the respective transcriptome data onto a curated interleukin and interferon interactome (Figure 6A–C).

These analyses further support significantly higher expression and specificity of the IL7R alpha gene and genes encoding for the IL12R heterodimer, i.e., L12RB1 and IL12RB2, in MCD. Most importantly, neither IL-7 nor IL-12 gene expression was found to be significantly upregulated in any of the conditions studied, thus suggesting paracrine or endocrine pathways or regulatory mechanisms other than transcriptional regulation within glomerular tissue. Considering interleukin biology, the former appears more likely since MCD’s pathological hallmarks do not support a significant presence of immune cell subsets commonly associated with IL-7 and IL-12 production in the target organ.

On the other hand, significant upregulation of the IL-33/IL-1RL1 ligand/receptor pair, suggesting possible autocrine activation signals relevant for MCD, was also observed. Based on the implication of IL-33 in diabetic kidney diseases [39], this observation warrants the inclusion of the IL-33/IL1RL1 axis in further investigations.

### 3.5. Immunohistochemistry and Immunofluorescence Staining for IL7R, IL12RB2 and Human AXL Tyrosine Kinase Receptor

Immunohistochemical and immunofluorescence methods were used to validate at the protein level and further investigate the observed increased expression of IL7R and IL12RB2 genes. As an additional protein, the expression of AXL tyrosine kinase, a well-known marker of renal inflammation [40] was also explored.

Consistent with gene expression data, phosphorylated IL7R alpha and IL12RB2 expression could be detected at the protein level by immunofluorescence or immunohistochemistry, respectively, in the same 3 of 11 MCD samples, but not in control glomeruli (Figure 7, panels A–D).

The remaining 7 of 11 MCD patients were negative or only very weakly positive. IL7R alpha synthesis in MCD was visible mostly along the capillary walls (Figure 7, panel B) and IL12RB2 production predominantly in podocytes (Figure 7, panel D) but not in controls, respectively (Figure 7, panels A and C).

AXL protein expression was detectable in 7 of 11 MCD samples along the capillary walls but not in controls (Figure 7, panels E and F). However, on the mRNA level, Axl was overabundant only in MN but not in MCD patients (data not shown).

## 4. Discussion

MCD pathogenesis is unclear. However, current consensus supports the main role of the immune system, with dysregulation of both T- and B-cells. Moreover, a variety of interleukins and inflammatory components have been reported to be involved.

Our study identifies patterns of altered glomerular gene expression shared by MCD and MN and an MCD-specific transcriptional profile of potential clinical significance.

In particular, we have observed an inflammatory scenario characterized by significant enrichment of GSs involved in IL-1 production and secretion, vascular cell activation, and, to some extent, those promoting granulocyte chemotaxis, in both MCD and MN glomeruli. These data are largely consistent with past studies, showing high expression of IL-1 and IL-1 receptors by podocytes and endothelial cells in both MN and MCD [41], and validate the integrity of our experimental technique.

Applying the same methodology, we observed an overexpression of genes associated with the response to IL-7 and IL-12, which appeared to represent a specific MCD not present in MN.

IL-7 is known to be produced by a variety of stromal cells and is of essential relevance in lymphoid cell homeostasis. Importantly, IL-7 production by human cultured podocytes has previously been reported [42]. Moreover, IL-7 receptor alpha was shown to be expressed by cultured human renal endothelial cells and in perivascular areas of human lupus nephritis kidney biopsies [43]. In addition, increased soluble IL-7R alpha in serum has been reported to reliably mirror the lupus nephritis clinical course [43]. The recent detection of anti-nephrin autoantibodies in MCD further highlights a possible link between the two disease entities [44].

More recently, podocytes simulation by IL-7 was shown to impair the glomerular filtration barrier, thus leading to nephrotic syndrome in adriamycin-treated rats. Moreover, IL-7 induced podocyte apoptosis with actin cytoskeletal disorganization and suppressed nephrin activation in experimental models [14].

Glomerular transcriptome data indeed indicate that genes involved in IL-7 receptor signaling are upregulated in MCD, as compared to MN and control samples. Notably, in a fraction of samples, expression of activated, e.g., phosphorylated, IL-7R alpha (CD127) protein was detectable by immunofluorescence staining. However, intriguingly, no evidence of increased expression of genes governing IL-7 production was observed, suggesting an ongoing modulation of glomerular responsiveness to a distantly produced cytokine.

IL-12 is classically produced by myeloid cells, including macrophages, dendritic cells and granulocytes, and plays key roles in Th1 polarization, NK-cell activation and IFN-gamma production [33,34]. However, IL-12 has previously been shown to be also produced by murine tubular cells in an autoimmune experimental model [45]. Moreover, in the past, human mesangial cells were reported to produce IL-12 in vitro [46]. However, similarly to IL-7, our data appear to suggest that responsiveness to, rather than production of IL-12, is augmented in MCD glomeruli.

Within this context, although IL-12 beta 1 low-affinity receptor expression in mesangial cells was previously reported [38], expression of the beta 2 chain, a key component of the high-affinity receptor, was not described thus far. Our data indicate that IL-12 beta 2 receptor expression is detectable at the gene and protein level in MCD glomeruli.

Two additional important aspects deserve special attention. First, GSs involved in the adaptive immune response, including its effector phase and somatic recombination of immune receptors, were significantly enriched in MCD compared to MN samples. Furthermore, in MCD glomeruli, we observed a concordant decrease in the expression of genes involved in cellular respiration and oxidative phosphorylation, compared with MN and control samples. This is typically detectable during active immune responses involving both the innate and adaptive immune systems characterized by increased engagement of the glycolytic metabolic pathway [47].

On the other hand, MCD glomeruli presented gene expression patterns resulting in the enrichment of GSs involved in the control of gene expression at the DNA and RNA level, as compared to both MN and control samples.

These data suggest that a systemic immune response might be responsible for the inflammatory scenario emerging from MCD glomerular transcriptome analysis, as further supported by the expression, at the protein level of AXL, of a typical kidney inflammation marker [48]. Notably, on the mRNA level, Axl was overabundant only in MN but not in MCD patients (data not shown). Nevertheless, on the protein level, Axl was overexpressed in three subjects of our MCD cohort.

Limitations of our study should be acknowledged. They include a relatively small number of samples and the use of mRNA from micro-dissected entire glomeruli. Further studies based on single-cell sequencing allowing the analysis of individual cellular components are warranted. However, our data confirm that micro-dissected samples from archival FFPE kidney tissue biopsies may reliably be used to investigate gene expression in different kidney disease groups. On the other hand, it has to be acknowledged that archival FFPE tissue biopsies are difficult to stain by immunofluorescence due to autofluorescence associated with the fixation process. Nevertheless, IL7RA positivity was clearly detectable in at least a fraction of MCD samples. Furthermore, subjects of the normal control group were significantly younger than the respective members of the two studied groups. Thus, we cannot exclude a confounding effect of age per se and features such as age-related effects on immune activation, even including increased IL-7 and IL-7R expression with age [49].

Based on these findings, IL7R and IL-12 antagonists may be of high interest in MCD therapy [50,51], particularly in adult patients, typically characterized by a delayed responsiveness to steroid treatment [2,3] and increased recurrency rates. Moreover, these treatments could likely spare patients the serious metabolic side effects associated with prolonged but frequently poorly effective steroid administration.

## 5. Conclusions

Our results demonstrate that archival FFPE-biopsies can be used to generate glomeruli-specific gene expression profiles suitable for systematic analysis of kidney-associated disease pathogenesis. Most importantly, they provide a data-driven rationale to experimentally address these MCD-specific features as biomarkers and novel drug targets. In this context, inhibiting activation of the IL-7 and the IL-12 pathways appears to be particularly promising.

## Figures and Tables

**Figure 1 biomedicines-11-00226-f001:**
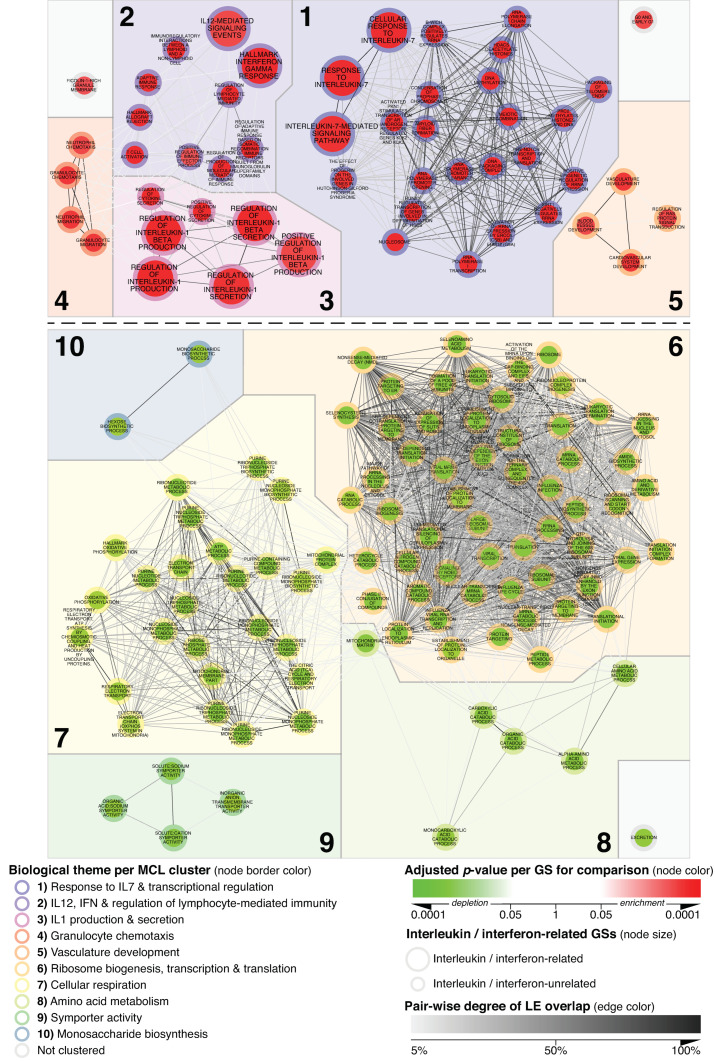
MCD’s glomerular transcriptional landscape delineating contextualized alterations in gene expression between MCD and N_CTRL. Any GS passing the significance thresholds of adjusted *p* < 0.001 and with at least 33.34% of its member genes comprised in its leading edge (LE) when comparing MCD vs. N_CTRL was mapped as a node within the basic network. Onto this scaffold, the parameters listed in the legend for the comparison of MCD vs. N_CTRL were mapped. By keeping all mapping parameters constant, this figure is directly comparable with Figure 2 and Figure 3.

**Figure 2 biomedicines-11-00226-f002:**
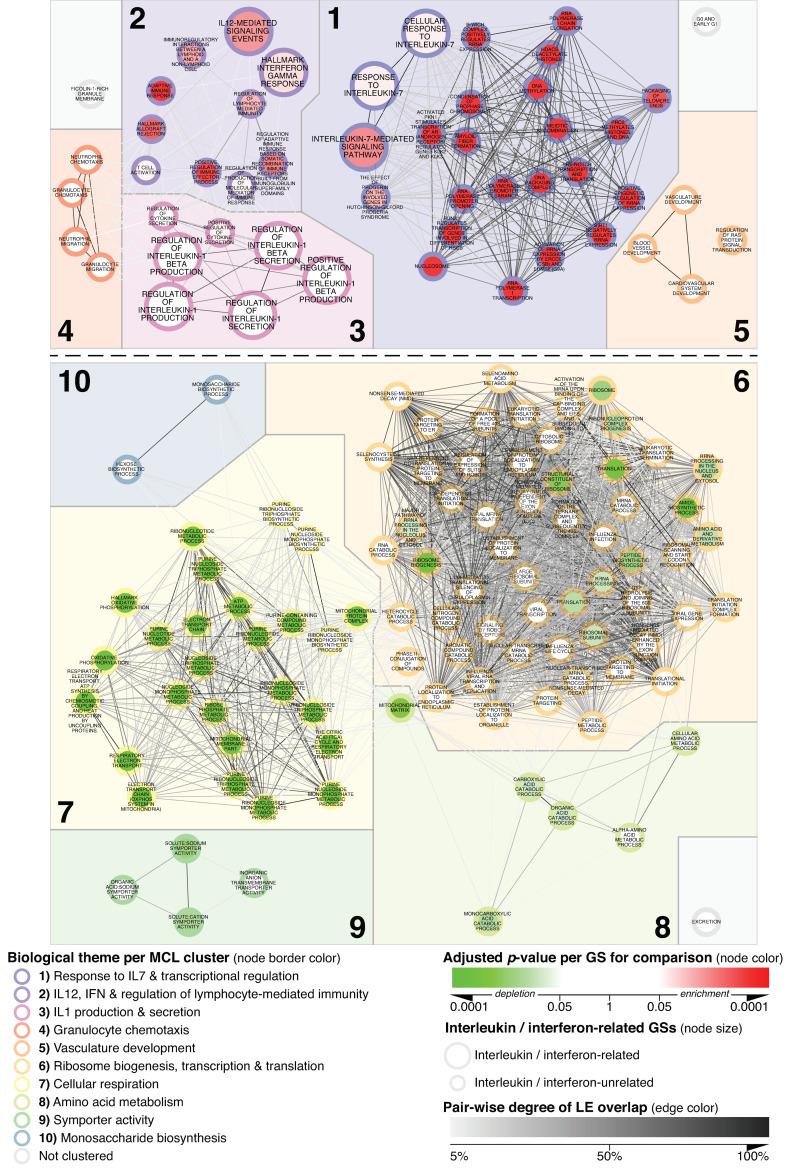
MCD’s glomerular transcriptional landscape delineating contextualized alterations in gene expression between MCD and MN. Any GS passing the significance thresholds of adjusted *p* < 0.001 and with at least 33.34% of its member genes comprised in its leading edge (LE) when comparing MCD vs. N_CTRL was mapped as a node within the basic network. Onto this scaffold, the parameters listed in the legend for the comparison of MCD vs. MN were mapped. By keeping all mapping parameters constant, this figure is directly comparable with Figure 1 and Figure 3.

**Figure 3 biomedicines-11-00226-f003:**
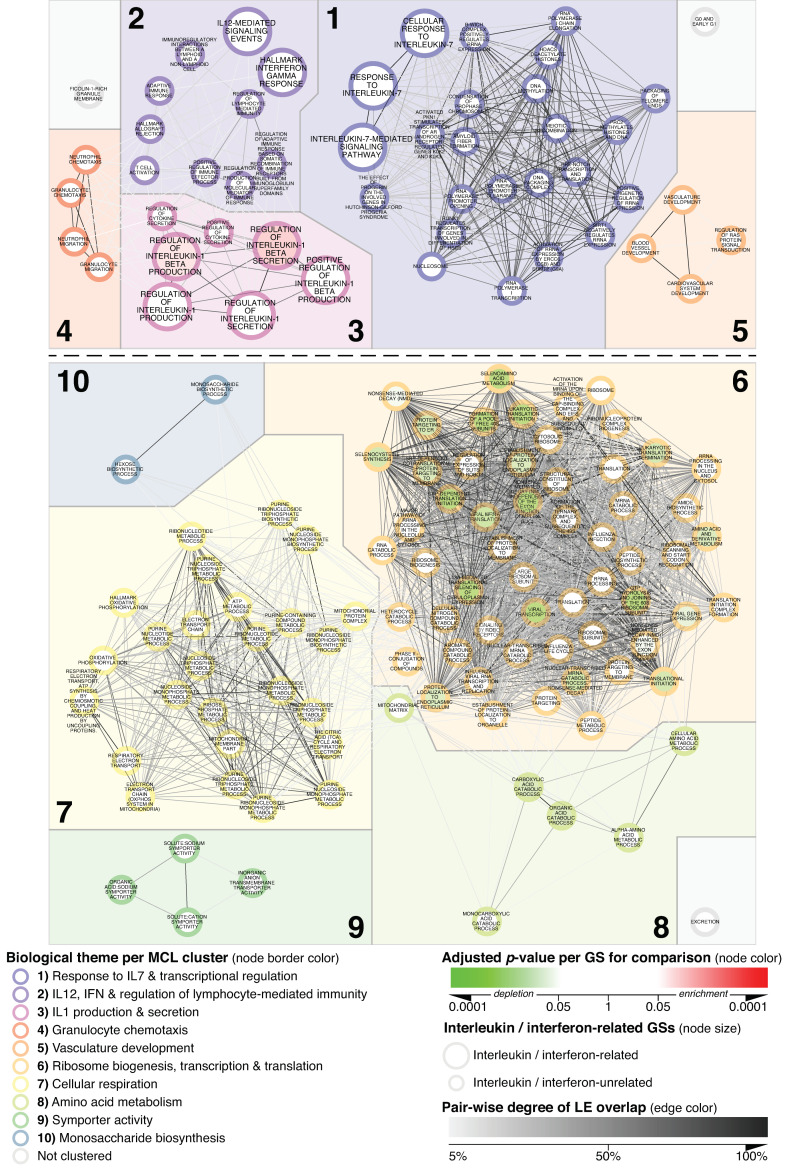
MCD’s glomerular transcriptional landscape delineating contextualized alterations in gene expression between MN and N_CTRL. Any GS passing the significance thresholds of adjusted *p* < 0.001 and with at least 33.34% of its member genes comprised in its leading edge (LE) when comparing MCD vs. N_CTRL was mapped as a node within the basic network. Onto this scaffold, the parameters listed in the legend for the comparison of MN vs. N_CTRL were mapped. By keeping all mapping parameters constant, this figure is directly comparable with Figure 1 and Figure 2.

**Figure 4 biomedicines-11-00226-f004:**
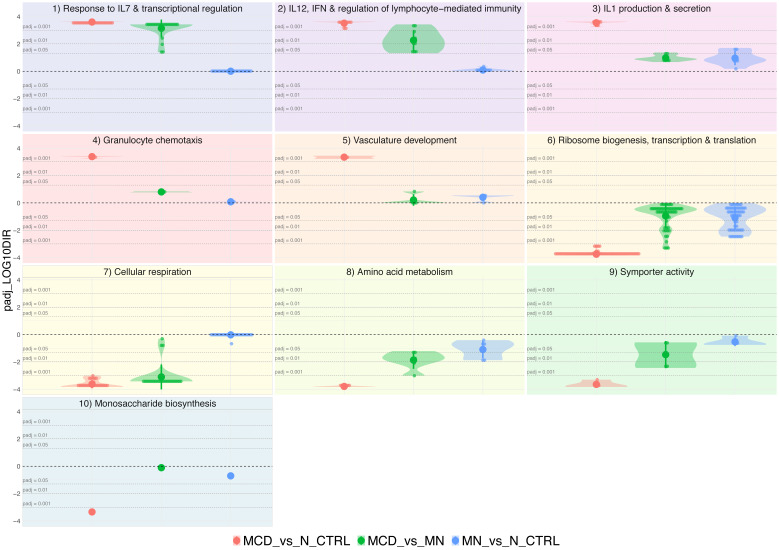
Violin plots displaying directional adjusted *p*-values for all comparisons per biological theme identified as significantly altered in MCD vs. N_CTRL. Adjusted *p*-values were log-transformed (log10), and the algebraic sign was assigned according to positive and negative enrichment signal, respectively. The dashed black horizontal line corresponds to an adjusted *p*-value 1, whereas the dashed grey horizontal lines indicate significance levels of adjusted *p*-values of 0.05, 0.01 and 0.001 for enrichment and depletion, respectively, as indicated.

**Figure 5 biomedicines-11-00226-f005:**
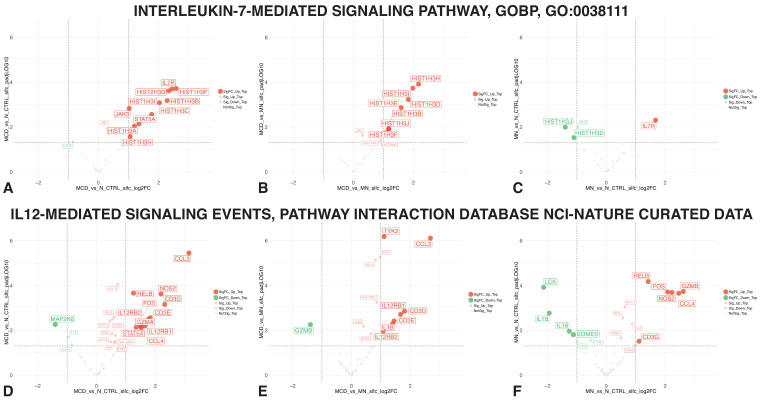
Volcano plots of IL-7 and IL-12-mediated signaling member genes. For the IL-7-mediated signaling pathway: (**A**) MCD vs. N_CTRL, (**B**) MCD vs. MN and (**C**) MN vs. N_CTRL. For the IL-12-mediated signaling pathway: (**D**) MCD vs. N_CTRL, (**E**) MCD vs. MN and (**F**) MN vs. N_CTRL. The volcano plots display all member genes of the respective pathway. Genes yielding adjusted *p*-values of <0.05 (above the horizontal dashed line) were labeled, and if they also passed the shrunk fold change criterion of >2 for upregulation or <−2 for downregulation (vertical dashed lines), they were highlighted further via increased size and saturation.

**Figure 6 biomedicines-11-00226-f006:**
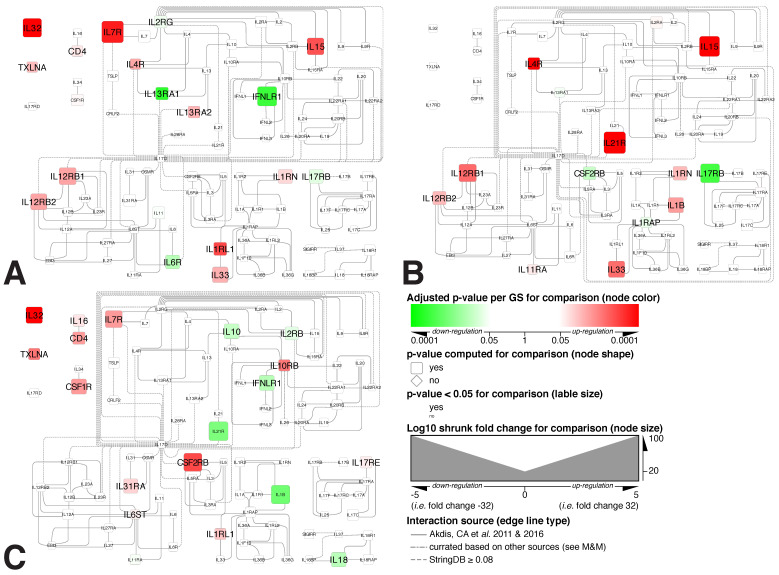
Alterations in interleukins-interferons ligand/receptor gene expression visualized within a curated interactome. (**A**) MCD vs. N_CTRL, (**B**) MCD vs. MN [33,34] and (**C**) MN vs. N_CTRL. The specific methodology used to construct this interactome is outlined in Section 2. The legend applies to all 3 panels.

**Figure 7 biomedicines-11-00226-f007:**
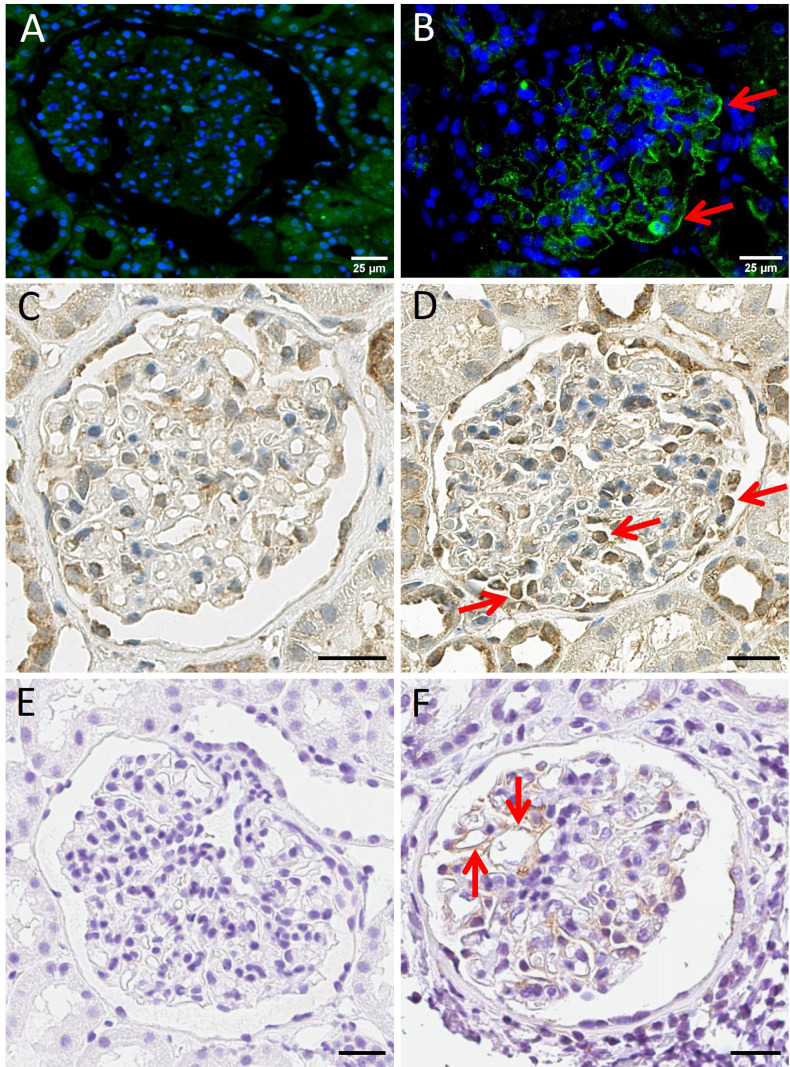
Immunofluorescence of IL7R alpha with normal control and MCD (panels **A**,**B**), and immunohistochemistry of IL12RB2 with normal control and MCD (panels **C**,**D**), as well as of AXL tyrosine kinase receptor normal control and MCD (panels **E**,**F**). Red arrows point toward positive glomerular areas in (**B**,**D**,**F**). MCD panels (**B**,**F**) show fine granular positivity along the capillary walls. Positivity in MCD panel (**D**) is mostly located in nuclear and perinuclear areas of podocytes. Control samples are either entirely negative (**A**,**E**) or show clearly less and possibly unspecific positivity (**C**). Scale bars in all images are 25 μm.

## Data Availability

Transcriptomic data have been submitted to the GEO repository. GEO accession number: GSE216841.

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
