# Peer review of "Network-Based Assessment of Minimal Change Disease Identifies Glomerular Response to IL-7 and IL-12 Pathways Activation as Innovative Treatment Target"

_biomedicines, 2023, doi:10.3390/biomedicines11010226_

Round 1

Reviewer 1 Report

Dear Editor,

I am attaching an extensive analysis of the manuscript which shows that the methods are poorly explained and not convincing. The results are cryptic and their biological significance is not understood.

For these reasons I must definitely reject the manuscript.

Best regards.

The abstract begins by talking about the treatment of MCD and the lack of response, especially in adults. So the reader would think that now we want to address the topic of resistance to treatment, perhaps related to markers. Then we change the subject and say that we want to elucidate the pathogenetic mechanisms, and then we don't understand the meaning of the introduction. Then we read that patients with NM were also taken, but NM had not been discussed previously. I advise the authors to make the abstract more understandable.

Unfortunately, FFPE samples are not ideal for studying RNA, but I would like that you to comment on this.

Were biological or technical replicates of the samples performed?

Had these patients undergone any corticosteroid treatment before the biopsy was performed?

Comment that the healthy were considerably younger than the sick. This could be a problem, also due to a certain level of basal inflammation that increases with increasing age. Please comment.

Line 96, do you mean ashr the package or the function?

Line 95, CPM do you mean counts per million? Please state it.

Row 98, it would be clearer if you write genes rather than features.

Row 98, 17561 is the number of genes differentially expressed among the three groups?

Rows 96-98, what threshold was chosen to declare that a gene is differentially expressed?

Rows 98-99, “Fast Gene Set Enrichment Analysis (fgsea) [23] to assess coordinated changes” what are coordinated genes? Do you mean they belong to the same pathway?

Rows 98-100, it is totally incomprehensible where the 26251 gene sets come from and what was done. Who determines the number of permutations? Is a gene set a collection of genes that participate in the same function or metabolic pathway? Finally on line 140 it is understood that a gene set is a pathway.

Placing graphics such as those in figure 1 in the manuscript makes no sense because they are completely illegible.

Row 153, “resolve redundancy” redundancy of what? Due to what?

Furthermore, "thereby summarizing enriched and depleted GSs pathways into 5 biological themes each", however, you decrease the resolution and increase the approximation. It is like considering an average rather than single values. I compressed but lost information. Comment or explain clearly.

By depletion and enrichment, do you mean that they were obtained by feeding down and up regulated genes, respectively?

Rows 155-158, couldn't IL-7 involvement be determined using single gene sets alone? Was there a need to derive biological themes?

Row 162, "In line with increased chemotaxis" but does this mean that there were also granulocytes in the sample analyzed?

Rows 162-164, explains this sentence better. Do you mean that chemotaxis is also not related to the activation of the immune system?

Row 175, "clusters" what are they? Gene set? Do you fear? Other?

Row 180, why is it intriguing?

I fear that having based oneself on the gene sets obtained from the comparison of MCD vs N_CTRL, and having used this network to compare MN may represent a strong bias.

I would have liked the authors to have shown the gene sets obtained from MN vs N_CTRL and those obtained from MCD vs MN. Do not map alterations within the MCD vs N_CTRL network.

What is Figure 2 for? What biological significance does it have? What is the use of seeing the distribution of p-adj of the individual GSs within a theme?

What is the biological information gained from figure 4?

Reviewer 2 Report

The authors have carried out expression studies in biopsies from patients with minimal change disease (MCD) and membranous nephropathy (MN) and compared them to control biopsies. They have identified changes in expression in pathways associated with IL-7 and Il-12 enriched in MCD biopsies, and others downregulated, when compared to controls and  suggest these warrant further investigation and are potentially useful for intervention studies. Similar findings were identified in MN biopsies although to a lesser extent. The study is generally well executed and written an have only minor comments. 

Minor Comments

1) The authors acknowledge that the age range of the controls differs significantly from that of the patients, indeed they are almost half the age of the patients. Given the apparent differences in response to treatment and incidences of relapse with age I think the authors should address this in a bit more detail, with references to potentially confounding effects of age and the immune response. For example Il-7 appears to increase with age-related disease (https://www.frontiersin.org/articles/10.3389/fimmu.2018.00586/full) and IL-7ra expression increases with age.

2) There should be some information on the treatments the patients were undergoing and a more detailed medical history e.g. comorbidities. Do any of the expression levels correlate with disease level?

3) Figure 1 is incredibly data dense and difficult to discern details. Would a heat map covering each pathway as a whole be easier to interpret, rather than the entire network? Or a summary table? The individual networks could then be shown as supplementary data, side-by side with columns for MCD vs control, MCD vs MN and MCD vs MN. the networks could then be viewed in more detail. As it is, the level of detail is so low it is effectively a heat map anyway.

4) Figure 2 is much clearer and easier to interpret at a glance. My only query is with panel 10 as it appears to be just dots rather than a violin plot. Is this a mistake or just a consequence of the data?

5) The authors attempted to confirm changes in expression at the protein level but only 3/11 MCD patients were determined as positive for phosphorylated IL-7ra and IL-12RB2 in 3/11 patients. Presumably these were the same 3 patients? It is not clear from the text. Were the others truly negative or ND because of high background, as highlighted in the discussion? These points should be clarified. 

6) Given that readers that are not renal specialists may read this journal the detail in Figure 5 needs to be improved and  the structures explained. The difference in expression between the control and the MCD biopsy is clear in panels C and D but it is not at all clear what the differences are. A judicious use of arrows and explanation is needed.

7)  The authors highlight the link between IL-7 and lupus nephritis in the discussion. They should also reference a recent paper identifying autoantibodies in MCD (https://jasn.asnjournals.org/content/33/1/238).

8) in Line 265 the authors state that 7/11 biopsies were positive for AXL but then in line 325 Axl expression was only overabundant in MN not MCD patients. Can the authors clarify what is going on.
